# Oral Administration System Based on Meloxicam Nanocrystals: Decreased Dose Due to High Bioavailability Attenuates Risk of Gastrointestinal Side Effects

**DOI:** 10.3390/pharmaceutics12040313

**Published:** 2020-04-01

**Authors:** Noriaki Nagai, Fumihiko Ogata, Hiroko Otake, Naohito Kawasaki

**Affiliations:** Faculty of Pharmacy, Kindai University, 3-4-1 Kowakae, Higashi-Osaka, Osaka 577-8502, Japan; ogata@phar.kindai.ac.jp (F.O.); hotake@phar.kindai.ac.jp (H.O.); kawasaki@phar.kindai.ac.jp (N.K.)

**Keywords:** meloxicam, nanocrystals, oral absorption, gastrointestinal lesion, drug delivery system

## Abstract

Meloxicam (MLX) is widely applied as a therapy for rheumatoid arthritis (RA); however, it takes far too long to reach its peak plasma concentration for a quick onset effect, and gastrointestinal toxicity has been observed in RA patients taking it. To solve these problems, we designed MLX solid nanoparticles (MLX-NPs) by the bead mill method and used them to prepare new oral formulations. The particle size of the MLX-NPs was approximately 20-180 nm, and they remained in the nano-size range for 1 month. The tmax of MLX-NPs was shorter than that of traditional MLX dispersions (MLX-TDs), and the intestinal penetration of MLX-NPs was significantly higher in comparison with MLX-TDs (*P* < 0.05). Caveolae-dependent endocytosis (CavME), clathrin-dependent endocytosis (CME), and micropinocytosis (MP) were found to be related to the high intestinal penetration of MLX-NPs. The area under the plasma MLX concentration-time curve (*AUC*) for MLX-NPs was 5-fold higher than that for MLX-TDs (*P* < 0.05), and the *AUC* in rats administered 0.05 mg/kg MLX-NPs were similar to rats administered the therapeutic dose of 0.2 mg/kg MLX-TDs. In addition, the anti-inflammatory effect of the MLX-NPs was also significantly higher than that of MLX-TDs at the corresponding dose (*P* < 0.05), and the therapeutic effect of 0.2 mg/kg MLX-TDs and 0.05 mg/kg MLX-NPs in adjuvant-induced arthritis (AA) rats showed no difference. Furthermore, the gastrointestinal lesions in AA rats treated repetitively with 0.05 mg/kg MLX-NPs were fewer than in rats receiving 0.2 mg/kg MLX-TDs (*P* < 0.05). In conclusion, we demonstrate that MLX solid nanoparticles allow a quick onset of therapeutic effect and that three endocytosis pathways, CavME, CME, and MP, are related to the high absorption of solid nanoparticles. In addition, we found that MLX solid nanoparticles make it possible to reduce the amount of orally administered drugs, and treatment with low doses of MLX-NPs allows RA therapy without intestinal ulcerogenic responses to MLX. These findings are useful for designing therapies for RA patients.

## 1. Introduction

Meloxicam (MLX) is categorized as a Class II drug in the Biopharmaceutical Classification System (BCS) [1] and is a selective inhibitor of cyclooxygenase-2. MLX is used in both human and veterinary medicines [2] and is widely applied for the relief of joint pain caused by rheumatoid arthritis (RA), osteoarthritis, and other chronic joint diseases [3]. Although gastrointestinal toxicity of MLX is observed in patients with RA, the rate and degree of onset are lower than with other non-steroidal anti-inflammatory drugs (NSAIDs). MLX is also a promising drug for the treatment of cancer and Alzheimer’s disease [4]. Despite this attractive pharmacological profile, the peak plasma concentration of MLX is reached only 3–7 h following the administration of an oral suspension, and 5–6 h after the administration of traditional MLX tablets [5,6]. This is far too long for a quick onset effect.

The permeability of dissolved MLX is a relatively good, and the bioavailability (BA) is approximately 89% after dissolution [7]. However, it has low dissolution and low solubility (approximately 4.4 μg/mL at water [1]), and these factors limit the absorption rate of MLX, since the undissolved meloxicam is little absorption. Thus, the slow oral absorption due to low solubility causes the slow expression of the pharmaceutical effects of MLX. In therapy for patients with acute exacerbation of rheumatism and osteoarthritis, rapid onset of pharmaceutical effects is important, and thus the development of a fast dispersible dosage form of highly soluble MLX nanocrystals (solid nanoparticles) is highly anticipated. In addition, the half-life of MLX is low (approximately 20 h) [8] in comparison with other NSAIDs [9,10]. Therefore, the development of a technique for controlled release is also important in the MLX category drugs, such as piroxicam and neloxicam.

Amorphous solid dispersion techniques [11], emulsification [12], nano-pulverization [13], and salt formation [14] have been studied as approaches to enhance the aqueous solubility of poorly soluble drugs by the many researchers. Nano-pulverization is one technique to prepare solid nanoparticles, and the reduction of particle size to the submicron range can lead to a higher BA and dissolution rate [15,16,17]. We also reported that the inclusion of 2-hydroxypropyl-β-cyclodextrin (HPβCD) with nanoparticle preparations increases in solid nanoparticles prepared by the bead mill method, as well as improving permeability through the cornea [18], skin [19,20], and small intestine [21] in rats and rabbits. These previous studies suggest that the method based on the bead mill with the addition of cyclodextrin is an effective and simple technique for improving the oral absorption and dissolution behavior of drugs. In addition, it is possible that a dose reduction following the enhancement of BA may lead to less direct irritation to the gastrointestinal area, and may attenuate the onset of gastrointestinal toxicity of NSAIDs.

In this study, we designed MLX nanocrystals (solid nanoparticles) for oral administration (MLX-NPs) and studied the mechanism of their transintestinal penetration. Moreover, we demonstrate the effect of an oral administration system based on MLX solid nanoparticles on the pharmacokinetics and onset of gastrointestinal side effects, and the therapeutic effectiveness of MLX-NPs for inflammation using model RA rats.

## 2. Materials and Methods

### 2.1. Reagents and Animals

All reagents used were of the highest purity commercially available. The caveolae-dependent endocytosis (CavME) inhibitor nystatin was obtained from Sigma-Aldrich Japan (Tokyo, Japan). MLX (4-hydroxy-2-methyl-N-(5-methyl-2-thiazoly)-2H-1,2-benzo-thiazine-3-carboxamide-1,1 dioxide), HEPES, phagocytosis inhibitor cytochalasin D, and Bayol F were purchased from Wako Pure Chemical Industries, Ltd. (Osaka, Japan). HPβCD was provided by Nihon Shokuhin Kako Co., Ltd. (Tokyo, Japan). Methylcellulose (MC) and heat-killed *Mycobacterium butyricum* were obtained from Nihon Shokuhin Kako Co., Ltd. (Tokyo, Japan) and Difco (Detroit, MI, USA), respectively. Streptomycin, penicillin, non-essential amino acid solution, L-glutamine, Dulbecco’s modified Eagle’s medium, and heat-inactivated fetal bovine serum were obtained from GIBCO (Tokyo, Japan). The Protein Assay Kit was purchased from Bio-Rad (Tokyo, Japan). The clathrin-dependent endocytosis (CME) inhibitor dynasore and micropinocytosis (MP) inhibitor rottlerin were provided by Nacalai Tesque (Kyoto, Japan). Transwell-Clear^TM^ plates were obtained from Costar (Cambridge, MA, USA). Inertsil^®^ ODS-3 column (diameter 2.1 mm) was purchased from GL Science Co., Inc. (Tokyo, Japan). Male 6-week old Dark Agouti (DA) rats were obtained from Shimizu Laboratory Supplies Co. Ltd. (Kyoto, Japan). The experiments using rats were carried out in accordance with the Animals Guidelines of both the Kindai University (KAPS-30-003, 1 April 2018) and the Japanese Pharmacological Society. For the induction of RA (adjuvant-induced arthritis rat, AA rat), 50 microliters of Bayol F containing 10 mg/mL heat-killed *Mycobacterium butyricum* was injected into the plantar region of the right hind foot and tail (vehicle, 50 µL of Bayol F was injected). The inflammation levels (paw edema) were evaluated by measuring paw volume by plethysmometry.

### 2.2. Preparation of MLX Solid Nanoparticle-based Oral Formulations

Nanoparticulation of MLX was carried out according to our previous reports [18,19,20,21]. HPβCD, MC, and MLX powder were added into 2 mL tubes containing 2 g of 0.1 mm-zirconia beads and dispersed in purified water. The mixtures were milled 30 times at 5500 rpm for 30 s at 4 °C by Micro Smash MS-100R (TOMY SEIKO Co., Ltd., Tokyo, Japan). The milled dispersions were filtered to remove the 0.1 mm-zirconia beads, and the filtrates dispersions were used as MLX-NPs. Traditional MLX dispersions (MLX-TDs), used as the control, were prepared by the addition of HPβCD, MC, and MLX powder into purified water (pH = 7). The prepared MLX-TDs and MLX-NPs were used to evaluate transintestinal penetration, BA, side effects, and therapeutic effects in this study. The compositions of meloxicam formulations prepared in this study were as follows: 0.5% meloxicam, 5% HPβCD, 0.5% MC. The formulations were diluted to 0.05 mg/mL and 0.2 mg/mL by using solution water containing 5% HPβCD and 0.5% MC, and used in each experiment. In the in vitro experiment using Caco-2 cell and rat intestine, the 0.2 mg/mL meloxicam was used, and the 0.05 mg/mL and 0.2 mg/mL meloxicam formulations were used in 0.05 mg/kg and 0.2 mg/kg administration, respectively, in the in vivo study using rats.

### 2.3. General Characteristics of MLX Formulations

The particle frequency, nanoparticle-number, atomic force microscope (AFM) image, zeta potential, crystal form, concentration, and solubility were measured as general characteristics in this study. The particle frequency and nanoparticle-number of MLX-NPs were determined using NANOSIGHT LM10 (QuantumDesign Japan, Tokyo, Japan). The measurement time, wavelength, and viscosity were set at 60 s, 405 nm, and 1.27 mPa⋅s, respectively. The particle frequency of MLX-TDs and MLX-NPs was also measured by SALD-7100 (Shimadzu Corp., Kyoto, Japan). The AFM images were obtained using an SPM-9700 (Shimadzu Corp., Kyoto, Japan), and the images were created by combining height and phase images. A micro-electrophoresis zeta potential analyzer model 502 was used to measure the zeta potential (Nihon Rufuto Co., Ltd, Tokyo, Japan). The crystal form was analyzed by a powder X-ray diffraction (XRD) method using Mini Flex II (Rigaku Co., Tokyo, Japan) under the following conditions: Diffraction angles, 5° to 90°; X-rays, 30 kV and 15 mA; scanning rate, 10 °/min. For these measurements of the crystal form, lyophilized MLX-NPs was used as the sample. The MLX concentration was determined by an HPLC method. An LC-20AT system (Shimadzu Corp., Kyoto, Japan) with an Inertsil^®^ ODS-3 column was used with a mobile phase consisting of potassium dihydrogen phosphate/methanol/acetonitrile (63/27/10, *v/v/v*%) at a flow rate of 0.25 mL/min. The MLX peak was detected at 7 minutes at a wavelength of 254 nm. In experiments to measure solubility, the MLX in solution and the MLX nanoparticles were separated by centrifugation at 100,000 g using an Optima^TM^ MAX-XP Ultracentrifuge (Beckman coulter, Osaka, Japan). The collected nanoparticles were dissolved in methanol. The MLX contents of the solutions and methanol-dissolved nanoparticles were measured by the HPLC method described above.

### 2.4. Transepithelial Penetration of MLX using Caco-2 Cell Monolayers

Ten percent heat-inactivated fetal bovine serum, 10 μg/mL streptomycin, 1000 IU/mL penicillin, 1% L-glutamine, and 1% non-essential amino acid were added into Dulbecco’s modified Eagle’s medium, and used to culture Caco-2 cells. 9 × 10^4^ Caco-2 cells/cm^2^ were seeded onto Transwell-Clear™, and cultured for 21 days. Transepithelial electrical resistance (TER) was measured using an epithelial Volt-Ohm meter (Millicell-ERS, EMD Millipore, Billerica, MA, USA). Caco-2 cell monolayers with TER values over 300 Ω⋅cm^2^ were used in the transepithelial penetration experiments. The monolayers were set in the reaction chamber with HBSS/HEPES solution (pH 7.4) in the basolateral side chamber and the MLX formulations on the apical side. Aliquots of the basolateral side solution were withdrawn at various times and replaced with the same volume of HBSS/HEPES solution. The MLX levels in the withdrawn aliquots were measured by the HPLC method described above. At the end of the experiment, the TER values were measured again to evaluate cell damage.

### 2.5. Intestinal Penetration of MLX using Rat Jejunum and Ileum

Eight-week-old rats were killed under deep isoflurane anesthesia, and the jejunum and ileum were carefully removed. Individual jejunum and ileum samples were set in methacrylate cells for intestinal penetration experiments. The donor and reservoir chambers were filled with MLX formulations and 10 mM HEPES buffer (pH 7.4), respectively. Samples of the solutions in the reservoir chamber were withdrawn with time to investigate the penetration of MLX nanoparticles, and replaced with the same volume of 10 mM HEPES buffer. The MLX levels were measured by the HPLC method described above. The experiments were carried out for 4 h under normal (37 °C) and low (4 °C) temperature conditions. At the low temperature, all energy-dependent endocytosis was inhibited. In experiments using endocytosis inhibitors, the inhibitors were added to the reservoir chamber 20 min prior to the addition of the MLX formulation to the donor chamber. The inhibitors chosen were nystatin (54 μM, CavME inhibitor) [22], dynasore (40 μM, CME inhibitor) [23], rottlerin (2 μM, MP inhibitor) [24], and cytochalasin D (10 μM, phagocytosis inhibitor) [22], and the trapezoidal rule up to the last measurement point (4 h) was used to analyze the area under the MLX concentration (*C*)-time curve (*AUC*_penetration_).
(1)$$AUC_{\mathrm{penetration}}=\int_{0h}^{4h}Cdt$$

### 2.6. Absorption of Orally Administered MLX

Cannulas were set into the right jugular veins of 8-week old rats under isoflurane anesthesia, and 200 µL samples of blood were collected from right jugular veins via cannulas at various times to evaluate the changes in MLX levels after the oral administration of 0.05 mg/kg or 0.2 mg/kg MLX formulations. The collected venous blood from fasted rats was centrifuged at 800 g for 20 min, and the MLX in the serum was extracted by the methanol, and measured by the HPLC method described above. The trapezoidal rule up to the last measurement point (24 h) was used to analyze the area under the MLX concentration-time curve (*AUC*_plasma_). The pharmacokinetics parameters were analyzed by Equations (2) and (3):(2)$$C_{\mathrm{MLX}}=C_{0}\times e^{-k_{e}\times t}$$
(3)$$C_{\mathrm{MLX}}=\frac{k_{a}\times F\times D}{V_{d}\left(k_{a}-k_{e}\right)}\left(e^{-k_{e}\left(t-\tau \right)}-e^{-k_{a}\left(t-\tau \right)}\right)$$

The distribution volume (*V*_d_) and elimination rate constant (*k*_e_) was calculated using Equation (1) and data (0–24 h, *t*) after a single injection (0.3 ml) of MLX solution in DMSO (0.04 mg/kg) into the femoral vein, and levels of 64.5 mL and 2.45 × 10^−2^·h^−1^, respectively, were obtained. The apparent absorption rate constant (*k*_a_) was estimated according to Equation (2). In the Equation, the *D*, *τ*, *F* show dose (µmol), lag time (h), and the fraction of MLX absorbed, respectively.

### 2.7. Content of MLX in Gastrointestinal Mucosa after the Oral Administration of MLX

Fasted rats were orally administered 0.2 mg/kg MLX formulations, and killed under deep isoflurane anesthesia. The stomach, jejunum, and ileum were carefully excised, washed in saline, and the mucosa collected. MLX in the mucosal tissues was extracted in methanol, and measured by the HPLC method described above. 

### 2.8. Lesions in the Gastrointestinal Mucosa after the Oral Administration of MLX

AA rats, 14–42 days after the injection of *Mycobacterium butyricum,* were orally administered 0.2 mg/kg MLX formulations twice a day (7:00 and 19:00) for 1 month, and then killed under deep isoflurane anesthesia. The stomach, jejunum, and ileum were carefully excised, washed in saline, and fixed in 10% formalin, and digital images were obtained. The lesion areas in the stomach, jejunum, and ileum were analyzed using Image J (NIH) and expressed as the percentage of the total area.

### 2.9. Anti-Inflammatory Effect of MLX Formulations in AA Rats

The anti-inflammatory effect was evaluated by changes in paw edema of AA rats orally administered 0.05 mg/kg or 0.2 mg/kg MLX formulations. Paw edema of the right hind foot was measured by plethysmometry, and expressed as the difference in paw volume of rats injected with or without *Mycobacterium butyricum*. The trapezoidal rule up to the last measurement point (42 days) was used to analyze the area under the MLX concentration-time curve (*AUC*_edema_).

### 2.10. Statistical Analysis

Student’s *t*-test and Dunnett’s multiple comparisons were used. *P* values less than 0.05 were considered significant. Data were represented as mean ± standard error (S.E).

## 3. Results and Discussion

### 3.1. Preparation of MLX Solid Nanoparticles for Oral Formulations

Two basic methods, bottom-up (precipitation techniques) and top-down (disintegration processes), were used to obtain drug solid nanoparticles [25]. Solid nanoparticles are built up from drug molecules dissolved in an organic solvent in the precipitation techniques [26,27]; however, the organic solvent may cause toxicity when its residue remains in the final product, making the use of organic solvents undesirable. On the other hand, in top-down processes, the raw material is broken down until nanosized particles are produced. Among the top-down processes, wet bead milling and high-pressure homogenization are the two most widely used methods. In particular, wet bead milling produces particles in a smaller size range in comparison with the high-pressure homogenization processes [28]. Thus, wet bead milling is suitable to prepare solid nanoparticles, although increasing aggregation occurs during storage, reducing the dissolution rate and absorption. Therefore, the prevention of aggregation is important in the design of nanocrystal formulations [29,30]. In our previous studies, we showed that HPβCD attenuated the aggregation of nanocrystals, probably by coating the particles with HPβCD [18,19,20,21]. In addition, we found that the presence of MC increased the mill efficiency of the bead mill method and prevented the separation of hydrophobic drugs and a polar solvent during the preparation of nanoparticles [18,19,20,21]. From these findings, we selected the bead milling method and used HPCD and MC to prepare MLX solid nanoparticles for this study. The particle size was decreased by the bead mill treatment (Appendix A) to yield MLX nanoparticles with a size in the range of 20 nm to 180 nm (Figure 1B–E) with a zeta potential of −15.5 mV (Figure 1G). Moreover, the solubility was enhanced by the bead mill treatment, with the solubility of MLX-NPs being 2.6-fold that of MLX-TDs (Figure 1F), and the XRD pattern of MLX was not changed by the bead mill treatment (Appendix A). It is known that the solubility of particles under 100 nm increases according to the Ostwald–Freundlich equation. As the particle size of MLX-NPs is 20 nm to 180 nm (Figure 1B–E), this may be the reason for the increase in solubility of MLX. In addition, our previous reports showed that the inclusion of HPβCD also enhanced the solubility of solid nanoparticles that were smaller than 200 nm [18,19,20,21], and this may also be related to the increase in MLX solubility. On the other hand, the Ostwald–Freundlich equation indicates that the dispersion stability decreases with the increase in size distribution. In this study, the MLX particles were slightly aggregated 1 month after preparation, although the particle size remained in the nano-range (Figure 2B–E). No degradation was observed for 1 month, and zeta potential remained unchanged 1 month after preparation (Figure 2F,G). These results suggest that HPβCD and MC are useful additions for the preparation of MLX nanoparticles.

### 3.2. Relationships between Energy-Dependent Endocytosis and the Intestinal Penetration of MLX Solid Nanoparticles from Orally Administered Formulations

Next, we investigated the mechanism of the intestinal penetration of MLX solid nanoparticles. In the clinic, the dose of MLX was 10 mg/day (max 15 mg/day) in the adult, and the body weight in adults was approximately 50 kg. Taken together, we selected a dose of 0.2 mg/kg (10 mg/50 kg) in this study. Caco-2 expresses several markers characteristic of normal small intestinal villus cells, and Caco-2 cell monolayers are widely used to predict intestinal drug penetration [31,32]. The TER of Caco-2 cell monolayers reflects cell growth and the formulation of tight junctions [33,34,35]. In this study, we used Caco-2 cell monolayers with TER over 300 Ω⋅cm^2^ (Appendix A). Both the trans-epithelial penetration and intracellular uptake of MLX-NPs were significantly higher than those of MLX-TDs (*P* < 0.05, Figure 3A,D). Moreover, MLX nanoparticles were detected on the basolateral side (Figure 3B,C, Appendix A), and no difference was observed in cell stimulation between MLX-TDs and MLX-NPs (Figure 3E). It was known that the width of the intercellular space was approximately 40–100 nm, and the particle size in MLX-NPs was 20–180 nm (Figure 1). These results show that the MXL nanoparticles in MXL-NPs can pass through Caco-2 cell monolayers in the solid-state (nanoparticles).

Next, we investigated the mechanism of the intestinal penetration of MLX nanoparticles using isolated rat intestine and various inhibitors of energy-dependent endocytosis (Figure 4).

High intestinal penetration was observed for MLX-NPs under normal conditions (37 °C), while intestinal penetration was prevented under low temperature (4 °C) conditions (*P* < 0.05, Figure 4A,B), and the decrease of intestinal penetration in MLX-NPs under low-temperature (4 °C) conditions were higher than that in MLX solution and MLX-TDs (Appendix A). It is known that energy-dependent uptake is inhibited under low-temperature conditions [36], thus energy-dependent uptake may be related to the intestinal penetration of MLX-NPs. Endocytosis is one form of energy-dependent uptake and has been reported to be involved in the uptake of nanoparticles into cells [18,21]. Therefore, we investigated MLX penetration of the jejunum and ileum in the presence of various endocytosis inhibitors. Endocytosis is mainly classified as CavME, CME, MP, and phagocytosis, which are inhibited by nystatin, dynasore, rottlerin, and cytochalasin D, respectively [22,23,24]. Nystatin binds to plasma membrane cholesterol, and dynasore blocks dynamin. Rottlerin inhibits fluid-phase endocytosis, and cytochalasin D prevents the disassembly and actin polymerization of the actin cytoskeleton. These actions cause the inhibition of each type of endocytosis and are used as selective inhibitors of endocytosis [22,23,24]. In both the jejunum and ileum, treatment with nystatin, dynasore, and rottlerin significantly inhibited the intestinal penetration of MLX (*P* < 0.05, Figure 4C–F). These results indicate that three endocytosis pathways (CavME, CME, and MP) are related to the uptake of MLX solid nanoparticles (Scheme 1). On the other hand, in contrast to the results with Caco-2 cell monolayers, no MLX nanoparticles were detected on the basolateral side of the jejunum and ileum. Therefore, it is hypothesized that the absorbed MLX solid nanoparticles are dissolved in the small intestine, and pass into the blood in the dissolved form (Scheme 1).

### 3.3. Usefulness of Oral Formulations of MLX Solid Nanoparticles as Therapy for RA Patients

It is important to evaluate the safety of MLX solid nanoparticles for application in orally-administered formulations. First, we compared the MLX absorption profiles of MLX-NPs and MLX-TDs. The intestinal penetration for MLX-NPs was significantly higher than for MLX-TDs (*P* < 0.05, Figure 5A), and the peak plasma concentration was reached in approximately 1.4 h following the application of MXL-NPs (Appendix A). Moreover, the apparent absorption rate constant (*k*_a_) of MLX-NPs was 7.6-fold that of MLX-TDs (*P* < 0.05, Appendix A). Hanft et al. and Dellgado et al. reported that traditional oral suspensions of MLX reach a peak plasma concentration 3–7 h after application, which was far too long for a quick onset of effect [5,6]. The use of MLX-NPs may improve these problems since the *t*_max_ of MLX-NPs is shorter than that of MLX-TDs (Figure 5A). In addition, the intestinal retention of MLX-NPs is significantly prolonged in comparison with MLX-TDs in the stomach, jejunum, and ileum (*P* < 0.05, Figure 5D–F). The enhanced solubility (Figure 1F), penetration via endocytosis (Scheme 1), and intestinal drug retention (Figure 5D–F) may result in an increase of MLX absorption from MLX-NPs (Figure 5A).

The plasma drug concentration reflects drug efficacy in therapy. Therefore, we attempted to obtain the same *AUC*_plasma_ between of MLX-TDs and MLX-NPs by measuring the plasma MLX concentration in rats receiving 0.2 mg/kg MLX-TDs or 0.05 mg/kg MLX-NPs, since the absolute BA in the MLX-MPs and MLX-NPs were 20.3 ± 4.9% and 91.9 ± 8.9%, respectively, and the *AUC*_plasma_ for MLX-NPs was approximately 5-fold that of MLX-NPs in Figure 5A. Consistent with this, the *AUC*_plasma_ values were similar between 0.2 mg/kg MLX-TDs and 0.05 mg/kg MLX-NPs (Figure 5B and C). Next, the therapeutic effect on inflammation was measured in AA rats (Figure 6). The therapeutic effect of MLX-NPs was significantly higher than that of MLX-TDs at the corresponding dose (*P* < 0.05), and the therapeutic effect (*AUC*_edema_) was similar between 0.2 mg/kg MLX-TDs and 0.05 mg/kg MLX-NPs. These results supported the data in Figure 5A, which is that the ratio of absorption for MLX-NPs is 5-fold higher than that for MLX-TDs.

The gastrointestinal ulcerogenic response is a known side effect of NSAIDs, and RA patients taking NSAIDs are more susceptible to NSAIDs induced gastric and small intestinal ulcerogenic lesions than other patients [37,38]. In this study, AA rats were used as the model of RA to investigate the side effects of MLX-TDs and MLX-NPs at corresponding *AUC*_plasma_ levels. Figure 7 shows the gastrointestinal lesions in AA rats after the repetitive administration of 0.2 mg/kg MLX-TDs or 0.05 mg/kg MLX-NPs for 1 month. Gastrointestinal lesions were observed following the repetitive oral administration of MLX-TDs. On the other hand, no intestinal lesions were detected in the rats treated with MLX-NPs, and there were fewer gastric lesions in rats treated with MLX-NPs than in rats administered MLX-TDs (*P* < 0.05). Previous studies have reported that NSAIDs inhibit cyclooxygenase and cause the depletion of endogenous prostaglandins (PG), resulting in the formation of gastric lesions [39,40,41]. In addition, the direct stimulation by NSAIDs in the stomach and small intestine induces the overexpression of neutrophils, the formation of nitric oxide (NO) via inducible NO synthase, and these changes in the levels of neutrophils, NO and prostaglandins cause the pathogenesis of intestinal ulceration [39,40,41]. Taken together, the data suggest decreasing the MLX dosage (the amount administered orally) by improving BA attenuates the direct stimulation by MLX in the stomach and small intestine, resulting in a decrease in the onset of gastrointestinal lesions. Further studies are needed to clarify the changes in neutrophils, NO, and prostaglandins in the stomach and small intestine of rats treated with MLX solid nanoparticles. The present results suggest that treatment with low doses of MLX-NPs may enable RA therapy without intestinal ulcerogenic responses to NSAIDs. On the other hand, the rat model is different from humans, since the BA of MLX is higher than that in the rat model [7]. Further study is needed in RA patients.

## 4. Conclusions

We designed formulations of MLX solid nanoparticles for oral administration. These MLX formulations show high rates of intestinal absorption and retention and enable a quick onset of the therapeutic effect. Moreover, we demonstrated that three energy-dependent endocytosis pathways (CavME, CME, and MP) are related to the high rate of intestinal penetration of MLX solid nanoparticles. In addition, we found that MLX solid nanoparticles make it possible to decrease the amount of MLX administered due to the improved BA, and that treatment with low doses of MLX-NPs enable RA therapy without the intestinal ulcerogenic responses to NSAIDs (Scheme 2). These findings are useful in studies to design therapies for RA patients.

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
