# Peer review of "Oral Administration System Based on Meloxicam Nanocrystals: Decreased Dose Due to High Bioavailability Attenuates Risk of Gastrointestinal Side Effects"

_pharmaceutics, 2020, doi:10.3390/pharmaceutics12040313_

Round 1
Reviewer 1 Report
The manuscript describes the oral bioavailability enhancement of meloxicam using a nanoparticle formulation There are some interesting aspects presented in this manuscript. But some questions and comments need to be addressed before acceptance for publication.
- Meloxicam product information and authors statement indicate that oral bioavailability is 89% in humans, with a standard formulation. Therefore, the 5-fold improvement reported in this manuscript is questionable. Is the rat model different from human? Is the control formulation used in this rat study different from the marketed product with regard to oral bioavailability?
- The concentrations of meloxicam and excipients in the formulations used for in vitro and in vivo studies need to be reported.
- Please discuss the width of intercellular space vs. the size of these nanoparticles, to address potential paracellular vs. transcellular absorption.
- Why is absolute bioavailability not reported, since an IV group of rats was apparently included?
- The evaluation of temperature dependence should also include meloxicam in solution or using the control formulation, to compare with the effect using the nanoparticle formulation.
Author Response
We carefully revised our manuscript according to the suggestions of the reviewer 1, and details are as follows.
< Q and A for Reviewer 1>
Q1. Meloxicam product information and authors statement indicate that oral bioavailability is 89% in humans, with a standard formulation. Therefore, the 5-fold improvement reported in this manuscript is questionable. Is the rat model different from human? Is the control formulation used in this rat study different from the marketed product with regard to oral bioavailability?
A1. The reviewer’s comment is correct. The high oral bioavailability is shown after the dissolution. However, the meloxicam is very slightly soluble, and the undissolved meloxicam is little absorption. Thus, the solubility limits the oral absorption (bioavailability) of meloxicam. In order to respond to the reviewer’s comment, we revised the sentence (line 50-51).
Q2. The concentrations of meloxicam and excipients in the formulations used for in vitro and in vivo studies need to be reported.
A2. The reviewer’s comments are very important. The compositions of meloxicam formulations prepared in this study were as follows: 0.5% meloxicam, 5% HPβCD, 0.5% MC. The formulations were diluted to 0.05 mg/mL and 0.2 mg/ml by using solution water containing 5% HPβCD and 0.5% MC, and used each experiments. In the in vitro experiment using Caco-2 cell and rat intestine, the 0.2 mg/mL meloxicam was used, and the 0.05 mg/mL and 0.2 mg/ml meloxicam formulations were used in 0.05 mg/kg and 0.2 mg/kg administration, respectively in the in vivo study using rats. In order to respond to the reviewer’s comment, we added these information (line 106-112).
Q3. Please discuss the width of intercellular space vs. the size of these nanoparticles, to address potential paracellular vs. transcellular absorption.
A3. Thank you very much for pointing this out. It was known that the width of intercellular space was approximately 40-100 nm, and the particle size in MLX-NPs was 20-180 nm (Fig. 1). Therefore, some nanoparticles may penetrate through the intercellular space. Otherwise, the intestinal penetration of MLX-NPs was remarkably suppressed under low temperature conditions, and at normal temperature by treatment with nystatin, dynasore or rottlerin (Fig. 4). From these results, it was suggested that both the penetration through intercellular space and three endocytosis pathways (CavME, CME and MP) are related the high bioavailability. In order to respond to the reviewer’s comment, we added these contents in the Results and Discussion (line 275-276).
Q4. Why is absolute bioavailability not reported, since an IV group of rats was apparently included?
A4. Thank you for pointing out this. The absolute bioavailability in the MLX-MPs and MLX-NPs was 20.3 ± 4.9% and 91.9 ± 8.9%, respectively. In order to respond to the reviewer’s comment, we added the data in the Results (line 358-359).
Q5. The evaluation of temperature dependence should also include meloxicam in solution or using the control formulation, to compare with the effect using the nanoparticle formulation.
A5. The reviewer’s comments are very important. In order to respond to the reviewer’s comment, we added the data in the Fig. S4 (line 304-305, Figure S4).
Thank you for great comments.

Reviewer 2 Report
Meloxicam (MLX) is widely applied as a therapy for rheumatoid arthritis (RA) and MLX is practically insoluble in water and exhibits a slow oral absorption and slow onset of action. gastrointestinal (GI) toxicity is often reported as drug-associated toxicity in RA patients. To enhance the therapeutic efficacy, while minimizing the side effects the authors developed MLX solid nanoparticles using the bead mill method. The manuscript looks good however, it did not have novelty in the MLX formulation development. Reconsider the manuscript after a major revision
Author Response
We carefully revised our manuscript according to the suggestions of the reviewer 2, and details are as follows.
< Q and A for Reviewer 2>
Q1. The manuscript looks good however, it did not have novelty in the MLX formulation development.
A1. The reviewer’s comments are very important. The solid MLX nanoparticles show high rates of intestinal absorption and retention, and enable a quick onset of the therapeutic effect. In addition, the MLX solid nanoparticles make it possible to decrease the amount of MLX administered due to the improved BA, and that treatment with low doses of MLX-NPs enable RA therapy without the intestinal ulcerogenic stimulation. We think that these findings are contain the novelty, and useful in studies to design the nanomedicine. Thank you very much for pointing this out.
Thank you for great comments.

Reviewer 3 Report
Add a note on significance level comparison in the abstract.
The half-54 life of MLX is low (approximately 20 h) [8] in comparison with other NSAIDs [9,10]. Therefore, the 55 development of a technique for controlled release is also important. But, this statement is not valid for all the NSAIDs. Applicable to piroxicam and neloxicam. Rewrite the statement with MLX category drugs.
What is quantity of MLX formulations used for the penetration study?
Include the mathematical formulas used for the calculation of intestinal permeation in ileum and jejunum.
What is the route of blood collection from the rats? Specify the 0.05 mg/kg or 0.2 mg/kg MLX formulations. Which formulation administered at 0.05 and 0.2 mg/kg MLX in rats?
Write the extraction method of MLX from rat serum.
What is reason for using 0.2 mg/kg MLX formulations for GI mucosal study. How to compare with 0.5 mg/kg MLX-NP.
In the abstract, 0.05 mg/kg MLX-NPs better than 0.2 mg/kg MLX-TDs. But, in methodology section used 0.2 mg/kg administration of MLX formulations. Which one is correct. Recheck the experimental section.
Authors never make a significance comparative statement in the results section throughout the manuscript. Always showed in figures. Update the results and discussion section with statistical significance in appropriate places in the text as well..
Author Response
We carefully revised our manuscript according to the suggestions of the reviewer 3, and details are as follows.
< Q and A for Reviewer 3>
Q1. Add a note on significance level comparison in the abstract.
A1. The reviewer’s comments are very important. In order to respond to the reviewer’s comment, we added a note on significance level comparison (P<0.05) in the abstract.
Q2. The half-life of MLX is low (approximately 20 h) [8] in comparison with other NSAIDs [9,10]. Therefore, the development of a technique for controlled release is also important. But, this statement is not valid for all the NSAIDs. Applicable to piroxicam and neloxicam. Rewrite the statement with MLX category drugs.
A2. The reviewer’s comment is correct. In order to respond to the reviewer’s comment, we rewrote the sentence to “the development of a technique for controlled release is also important in the MLX category drugs, such as piroxicam and neloxicam” (line 57).
Q3. What is quantity of MLX formulations used for the penetration study?
A3. Thank you very much for pointing this out. The compositions of meloxicam formulations prepared in this study were as follows: 0.5% meloxicam, 5% HPβCD, 0.5% MC. The formulations were diluted to 0.05 mg/mL and 0.2 mg/ml by using solution water containing 5% HPβCD and 0.5% MC, and used each experiments. In the in vitro experiment using Caco-2 cell and rat intestine, the 0.2 mg/mL meloxicam was used, and the 0.05 mg/mL and 0.2 mg/ml meloxicam formulations were used in 0.05 mg/kg and 0.2 mg/kg administration, respectively in the in vivo study using rats. In order to respond to the reviewer’s comment, we added these information (line 106-112, 280, 293).
Q4. Include the mathematical formulas used for the calculation of intestinal permeation in ileum and jejunum.
A4. Thank you for pointing out this. The trapezoidal rule up to the last measurement point (4 h) was used to analyze the area under the MLX concentration-time curve (AUCpenetration). In order to respond to the reviewer’s comment, we added the equation (Eq. 1) (line 165).
Q5. What is the route of blood collection from the rats? Specify the 0.05 mg/kg or 0.2 mg/kg MLX formulations. Which formulation administered at 0.05 and 0.2 mg/kg MLX in rats?
A5. The reviewer’s comments are very important. The blood was collected from right jugular vein of rats administered with MLX. The formulation containing 0.5% MLX was diluted to 0.05 mg/mL and 0.2 mg/ml by using solution water containing 5% HPβCD and 0.5% MC, and used each experiments, and in the in vivo study, the 0.05 mg/mL and 0.2 mg/ml MLX formulations were used in 0.05 mg/kg and 0.2 mg/kg administration, respectively. In order to respond to the reviewer’s comment, we added these information (line 106-112, 169).
Q6. Write the extraction method of MLX from rat serum.
A6. Thank you for pointing out this. The MLX in the serum were extracted by the methanol. In order to respond to the reviewer’s comment, we added the information in the Materials and Methods (line 172).
Q7. What is reason for using 0.2 mg/kg MLX formulations for GI mucosal study. How to compare with 0.5 mg/kg MLX-NP.
A7. Thank you very much for pointing this out. In the clinic, the dose of MLX was 10 mg/day (max 15 mg/day) in the adult, and the body weight in adults were approximately 50 kg. Taken together, we selected the dose 0.2 mg/kg (10 mg/50 kg), and the dose in 0.5 mg/kg were determined according to the data of Fig. 5A and B in this study. In order to respond to the reviewer’s comment, we added the contents in the Results and Discussion (line 266-268).
Q8. In the abstract, 0.05 mg/kg MLX-NPs better than 0.2 mg/kg MLX-TDs. But, in methodology section used 0.2 mg/kg administration of MLX formulations. Which one is correct. Recheck the experimental section.
A8. The reviewer’s comments are very important. The MLX absorption levels and anti-inflammatory effect were similar to between of 0.05 mg/kg MLX-NPs an 0.2 mg/kg MLX-TDs. In the contrast to the results with drug absorption and therapeutic effect, the gastrointestinal lesions in AA rats treated repetitively with 0.05 mg/kg MLX-NPs were fewer than in rats receiving 0.2 mg/kg MLX-TDs. In order to respond to the reviewer’s comment, we recheck the contents in the manuscript.
Q9. Authors never make a significance comparative statement in the results section throughout the manuscript. Always showed in figures. Update the results and discussion section with statistical significance in appropriate places in the text as well.
A9. Thank you very much for pointing this out. We added a note on significance level comparison (P<0.05) in the results and discussion.
Thank you for great comments.

Round 2
Reviewer 1 Report
It is important to clearly state in the discussion that meloxicam product information indicates that oral bioavailability is 89% in humans, with a standard formulation. Therefore, the 5-fold improvement reported in this manuscript in rat may not translate to human. This could be because the rat model is different from human, or the control formulation used in this rat study different from the marketed product with regard to oral bioavailability.
Line 14 should read ...too long... not ...to long...
Author Response
< Q and A for Reviewer 1>
Q1. It is important to clearly state in the discussion that meloxicam product information indicates that oral bioavailability is 89% in humans, with a standard formulation. Therefore, the 5-fold improvement reported in this manuscript in rat may not translate to human. This could be because the rat model is different from human, or the control formulation used in this rat study different from the marketed product with regard to oral bioavailability.
A1. Thank you very much for pointing this out. In order to respond to the reviewer’s comment, we mentioned that “the rat model is different from human” in the Discussion. Thank you for pointing out this (line 397-398).
Q2. Line 14 should read ...too long... not ...to long...
A2. The reviewer’s comments are very important. In order to respond to the reviewer’s comment, we corrected to “...too long...” (line 14).
Thank you for great comments.

Reviewer 2 Report
Accept in the present form
Author Response
< Q and A for Reviewer 2>
Q1. Accept in the present form.
A1. Thank you for great comment.
